# Psychophysical Health Factors and Its Correlations in Elderly Wheelchair Users Who Live in Nursing Homes

**DOI:** 10.3390/ijerph17051706

**Published:** 2020-03-05

**Authors:** Natalia Wołoszyn, Joanna Grzegorczyk, Agnieszka Wiśniowska-Szurlej, Justyna Kilian, Andrzej Kwolek

**Affiliations:** Institute of Health Sciences, Medical College of Rzeszow University, 35-310 Rzeszow, Poland; joannag@pro.onet.pl (J.G.); wisniowska@vp.pl (A.W.-S.); justynakilian110@gmail.com (J.K.); kwoleka@o2.pl (A.K.)

**Keywords:** elderly, physical health, mental health, wheelchair user, nursing homes

## Abstract

*Background:* The aging is a multi-faceted process comprising both—the physical and mental alterations. Thus, the aim of the study was to evaluate the variables affecting the psychophysical state of the elderly people using wheelchairs and living in nursing homes (NH). *Methods:* 165 older wheelchair users were included in the study after meeting the inclusion criteria and expressing written consent. The assessment involved cognitive functioning, depression, body balance and flexibility, lung capacity and upper limbs dexterity, strenght and endurance. *The results* showed negative correlation between depression and balance, upper limb dexterity and endurance and shoulder flexion. A positive correlation between cognitive functioning and balance, upper limb dexterity, strenght and endurance, lung capacity and joint mobility was determined. Also, the study proved positive correlation between daily functioning and functional fitness, muscle strength and endurance, body flexibility, joint mobility. The regression analysis showed that better scores in balance test and joint mobility implied with higher scores in cognitive functioning. The most important determinants of functional fitness were balance, hand grip strenght and joint mobility. *Conclusion:* The future study should be focused on developing interventions aimed at senior wheelchair users living in NHs to prevent the deterioration of their mental and physical fitness.

## 1. Introduction

It is expected that the number of elderly people in the world will amount to 1.4 bill in 2030, and 2.1 bill in 2050. In 2100 this number may increase up to the level of 3.1 bill [1]. By the end of 2018 the number of people over 60 in Poland was 9.5 mil and 106,000 persons in this group were under institutional care [2,3].

Ageing is a multifaceted process involving both physical and psychological changes. Physical and mental fitness allows for independent functioning and as such it is fundamental for maintaining independence in the daily life [4]. Ageing involves the decrease of muscle strength, which entails loss of independence in everyday life [5]. Muscle strength of lower limbs is crucial for the ability to move around while an appropriate strength and balance of upper body are essential for keeping the ability to dress oneself, keep one’s face and hands clean and perform light, domestic chores [6]. At first, elderly people have difficulties with carrying out the activities which require strength, balance and manual dexterity (heavy, domestic chores, long walks, stairs climbing, clipping toenails, shopping). Seniors remain independent the longest in the scope in which it involves fitness of upper body (eating meals, maintaining basic hygiene) [7,8,9]. In their meta-analysis Vermeulen et al. presented a number of factors decreasing the level of the daily functioning and suggesting the monitoring of disability indicators in seniors in order to restore or maintain the independence of seniors in their everyday life [10].

The disability resulting from chronic diseases, decrease of muscle strength, fear of falling and progressive mobility disorders has impact on the increase of the percentage of the elderly people living in Nursing Homes (NHs) who use wheelchairs. It is estimated that the percentage of elderly, wheelchair users in NHs is approx. at the level of 25% [11,12]. Extensive use of wheelchairs limits physical activity, which—in turn—leads to decrease of disability and increase of dependency on others [13]. Restricted mobility may make social interactions difficult, and this translates into increased risk of depression [14].

Symptoms of depression and cognitive impairments are very common in elderly people living in NHs, and the percentage of persons with mood disorders there is at the level of 81%, which is connected, among others, with low level of functional fitness and limited mobility [15].

It needs to be noted that the number of seniors with reduced cognitive abilities fluctuates between 25% to 50% [16]. According to Drageste the very move to the NH causes stress associated with change of residence, adaptation to new living conditions, change of habits and new people. Such changes may contribute to the development of depressive disorders [17]. Depression in elderly people is often associated with disability. However, studies examining the link between depression and functional fitness in NH residents diverge. Murphy proved strong correlation between mood disorders and increased disability [18]. Nakamura et al. evidenced that symptoms of depression are strong predictors of becoming dependent on other people in the future, and these correlations escalated in case of persons living in NHs [19]. The authors believe that depression has impact on the decrease of self-motivation and initiative, which translates into limited physical activity and decreased functional fitness [19]. On the other hand, Chung and Condersson et al. did not identify any correlation between functional fitness and symptoms of depression in seniors living in NHs [20,21]. During the observations of seniors Makizako et al. identified decrease in daily functioning together with cognitive impairment and worsening of depression [22].

Decrease of cognitive abilities is a process that accompanies the process of ageing and it is subject to numerous changes which may cause various psychological and physical illnesses in seniors [23]. Additionally, growing number of studies show that there is a connection between the disability and the impairment of cognitive functions and subsequent development of mild cognitive impairment (Mild Cognitive Impairment, MCI) or senile stupor [24,25,26]. Boyle et al. demonstrated that decreased physical fitness is associated with increased risk of development of cognitive impairments in elderly people [27]. Similarly, Auyeung et al. showed that all results from measuring physical fitness in the four-year observation process entailed decreased cognitive abilities in elderly people [28].

The overview of the literature of the subject allows the authors to claim that there are few studies focusing on elderly people who are wheelchair users and live in NHs in Poland, although the correlation between physical and mental health is a well-established fact. Thus, the authors made it the aim of the study to assess the variables that affect psychophysical condition of the elderly people using wheelchairs who live in the south-east Poland.

## 2. Materials and Methods

### 2.1. Study Participants

This was a cross-section study conducted in six, randomly chosen NHs for elderly people and for people with chronic, somatic diseases in the south-eastern Poland. Before the study was started the managing authorities of NHs were informed about the purpose and about the course of the activities to be carried out within the framework of the research. Their consent was a prerequisite to start the relevant actions.

In order to be qualified for the study the participants had to meet the following criteria: (1) older adults ≥ 65 years old; (2) residents at the facility ≥ 3 months; (3) reliant on wheelchairs to move around and (4) consent for the study. The exclusion criteria were as follows: (1) severe and acute cardiovascular, musculoskeletal, or pulmonary illnesses (2) suffering from spinal cord injury with no rehabilitation potential (3) lack of consent of the elderly person or their doctor for participation in the exercises.

615 residents of NHs were randomly chosen as candidates for the study. After having applied the exclusion criteria 165 persons from among the randomly chosen 615 candidates were qualified for the study. The flow diagram shows the relevant data with focus on the participant selection and dropout percentage (Figure 1).

In line with the provisions set forth in the Declaration of Helsinki the seniors were informed about the purpose of the study and the course of activities to be conducted within the framework of the study. They gave their consent to participate in the research project. This project was approved of by the Bioethics Committee at the University of Rzeszów and was assigned the number 9/11/2017.

### 2.2. Sample Size

In total, there are 1092 persons over 60 years old who live in NHs in the Podkarpackie province [2] 25% of those are elderly wheelchair-dependent people [12]. The sample size was calculated using the Online Sample Size Calculator. The calculation is based on the following assumptions: the confidence level of 95% (0.95) and fraction size at the level of 0.5 with a maximum estimation error of 5%.

### 2.3. Research Procedure

The research project is a team-work and it is divided into two stages. The first stage of the project involved conducting interviews and collecting the socio-demographic data. The second stage covered performance of the anthropometric measurements and carrying out functional tests. The study covered the period of 2 weeks and it consisted in observing the daily functioning of the participants at the facilities where they had been living and the level of their depression. The objective at this stage was to gain better understanding of the participants’ daily performance.

#### 2.3.1. Cognitive Functioning

Cognitive functioning was assessed using the Mini-Mental State Exam, MMSE, which is a brief, quantitative measure of cognitive status of adults. The maximum score is 30. Scores at the level of ≥ 24 indicates no cognitive impairment, scores of 19–23 point to mild cognitive impairment, scores of 10–18 show moderate cognitive impairment, and scores of 0–9 mean severe cognitive impairment [29].

#### 2.3.2. Depression

Depression was assessed using the Geriatric Depression Scale, GDS, which was composed of 15 questions. This scale allows for subjective assessment of the mood, sense of happiness or lack thereof. A score over 5 points indicates mood disorders. The higher the score the more intense the symptoms of a depression are. The maximum score is 15. Lower scores are to be interpreted in the following way: score of 11–15 points indicates severe depression, score of 6–10 points—depression of a moderate degree, and score of 0–5 points—no depression [30].

#### 2.3.3. Daily Functioning

Daily functioning was assessed using the Barthel Index, BI. This index determines the degree of independence in terms of self-sufficiency, mobility, personal hygiene and control over sphincters. The maximum score is 100 points. The interpretation of the individual scores follows the description of the ranges they fit in 91–100—independent; 62–85—dependent; 21–6—almost completely dependent; 0–20—completely dependent [31].

#### 2.3.4. Body Balance Assessment

The Berg Balance Scale, BBS is a tool for examining elderly people with static and dynamic imbalance. Participants were asked to perform 14 types of predetermined movements in a standing and sitting position. A maximum score of 56 points is possible here. The score below 20 points indicates high risk of fall and need of using wheelchair by the participants [32].

#### 2.3.5. Body Flexibility

Upper body flexibility was assessed by using the Back Stretch Test, BS and lower body flexibility was assessed through the Chair Sit and Reach Test, CSR. The BS test requires the participant to bring together the fingers of his both hands together behind the back, with one arm placed over the upper body, and the other arm behind the lower back. Then, the distance between the fingertips of both hands is measured. The score is given in centimeters and it is positive if the fingers overlap or negative—if the fingers do not touch. The CSR test requires the participant to reach forward with his fingers towards the toes of his/her extended leg while sitting on a chair, the knee and the back being straight. Then, the distance between the fingertips and the tips of the toes is measured. The score in centimeters is an indicator here and it is positive when fingers and toes overlap, and it is negative in case the fingertips do not touch the toes [33].

#### 2.3.6. Muscle Strength and Endurance

Hand grip strength, HGS was carried out with the use of a hand dynamometer (Jamar Plus + Digital Hand Dynamometer, Patterson Medical). According to the guidelines of the American Society of Hand Therapists the measurement was performed in the sitting position with arms adducted, not rotated and elbows bent at 90 degrees. The dynamometer was held in the dominant or uninjured hand. Participants were then asked to grip the dynamometer as hard as possible. The result was read and recorded. The unit of measurement was kilogram. Upper limb muscle endurance was measured with the use of 30-s Arm Curl Test, ACT. The consists in doing as many arm curls with a handle as possible in 30 s, in the sitting position. A handle weighing 2 kg for women or 3.5 kg for men was held in the dominant hand. The number of repetitions is the indicator of the test [34].

#### 2.3.7. Joint Mobility

Shoulder mobility (shoulder flexion, extension, abduction) was assessed by the participant performing active movements in the sitting position. The measurements were made using standard goniometer [35].

#### 2.3.8. Manual Dexterity of the Upper Limb

Test Box and Blocks (Box & Block Test, Performance Health International Ltd., Sutton-in-Ashfield, UK), BBT is tool for examining upper-limb function. In the BBT test one hundred and fifty 2.5 cm wooden blocks in many different orientations are placed on the side of the partition with the testing hand. A subject’s score is equal to the number of the blocks transported over a partition in one minute. The subject can select blocks in any order to transport them over the partition as quickly as possible, with the only requirement being that the subject’s fingertips cross the vertical plane of the partition [36].

#### 2.3.9. Lung Capacity (PEF, FEV1)

Peak expiratory flow (PEF) and forced expiratory volume in 1 s, FEV1. Lung Capacity was measured by peak flow meter (TruZone Peak Flow Meter, Trudell Medical International, Trudell Medical International, London, ON, Canada). Participants were asked to take a deep breath and blow air into the meter as fast as they could [37].

### 2.4. Statistical Analysis

The first step involved calculating the percentage values for the individual variables on the nominal and ordinal scales. Measures of central tendency, mean, measure of dispersion and standard deviations were calculated for the quantitative variables. Next stages of the project consisted in performing *Spearman rho* correlations between the variables which featured at least the ordinal level of measurement. After having conducted the parametric tests, linear, stepwise regression analysis was conducted with the aim to verify the models. The analysis was conducted with the use of IBM SPSS 25.0 statistical package (SPSS Inc., Chicago, IL, USA). The statistical significance was accepted at the level *p* < 0.05.

## 3. Results

The study covered 165 elderly persons, including 98 (59.4%) women and 67 man (40.6%). The average age of the elderly person was 74.35 ± 7.33 years. Almost 50% of the participants were widows or widowers (*n* = 73; 44.2%), and 60% of the participants were secondary school graduates. Additionally, the most common chronic illnesses included: cardiovascular diseases (*n* = 117; 70.9%) and neurological diseases (*n* = 113; 68.5%). All seniors were living in NHs for at least 4 months and were wheelchair users for at least 4 months. The average MMSE score was 24.94 ± 3.37 points. One third of the residents had mild dementia (36.97%), slightly lower percentage of persons suffered from cognitive impairment without dementia (27.27%). Over a half of the participants had moderate depression (*n* = 92; 55.76%). The average daily functioning score of the participants was evidenced at the level of 54.36 ± 11.78. The largest group was made by persons with severe dependence (*n* = 113; 68.48%). Other participants were moderately dependent (*n* = 52; 31.51%). There were differences between sexes, age, BMI, marital status, daily functioning, manual dexterity of the upper limb, muscle strength and endurance, shoulder flexion and lung capacity. Basic sociodemographic data and functional fitness parameters are shown in Table 1.

The results of the tests performed testify to the negative correlation between depression, BI, BBS, BBT, ACT and shoulder flexion. Furthermore, no correlation was determined between HGS, upper lower body flexibility, extension and abduction of shoulder as well as lung capacity. Moreover, negative correlation between BP and depression among men and between BBT, ACT and depression among women was noticed.

Positive correlation was identified between cognitive functioning and BBS, BBT, BS, muscle strength and endurance, shoulder flexion and abduction as well as PEF. No correlation between mental health and CSR, shoulder flexion and FEV1 was determined. In addition, men manifested positive impact of BI, BBS, BBT, PEF, muscle strenght and endurance, shoulder flexion and extension on cognitive functioning, while data analysis for women revealed positive impact of BBS, HGS, ACT, shoulder abduction and BBT on cognitive functioning.

The level of daily functioning correlates positively with balance, manual dexterity, muscle strength and endurance, body flexibility, joint mobility and FEV1. The correlation between daily functioning and lung capacity was found neither among women nor men (Table 2).

The following variables: BI, BBS, BBT, BP, CSR, HGS, ACT, shoulder flexion, extension and abduction, PEF and FEV1 were entered in the stepwise regression analysis. The regression analysis showed that increased level of cognitive functioning was associated with better dynamic balance, and in this respect the study was operationalized by means of BBS test and shoulder flexion. In the population under study it was also shown that the symptoms of depression build up when functional and strength of the upper limb decrease. Higher level of daily functioning was contingent on the static and dynamic balance, flexibility of the upper body, grip strength and shoulder flexion (Table 3).

## 4. Discussion

The study results proved higher functional disability, lower manual dexterity, muscle strenght and lung capacity among women comparing to men. Positive impact of BBS, BBT, HGS and ACT on cognitive functioning was determined for both women and men. Furthermore, the daily functioning was positively correlated with balance, manual dexterity, body flexibility, muscle strength and endurance and joint mobility. Body balance and shoulder flexion were also the important factors affecting the psychophysical health condition of older NHs inhabitants.

The study covered elderly people who were wheelchair users. The residents of NHs make a heterogeneous group in terms of physical and mental health [38]. Decreased physical fitness and dependence on caretakers which are typical for elderly people as well as somatic symptom disorders are the main risk factors for the development of depression. These circumstances also imply poor mental health, quality of life and the well-being [39].

The number and the type of chronic diseases contribute to the decreased physical and mental fitness. 70% of the elderly people suffer from cardiovascular diseases. Marginally lower percentage of the participants had neurological illnesses (68.5%). Over half of the participants were diagnosed with musculoskeletal diseases. It was demonstrated that chronic diseases are attributable to progressing functional disability in elderly people. The occurrence of one disease increases significantly the risk of another illness which—in turn is associated with decreased physical and mental functions [40]. Sung found out that 75% of the women who took part in a study were suffering from at least one of the above–mentioned chronic diseases, which translated into low physical fitness and dependence on others in daily life [41]. Furthermore, Welmer et al. showed that chronic diseases were firmly connected with physical disability, and this correlation was particularly strong in seniors in their sixties and seventies [42]. It was evidenced that additional chronic diseases and wheelchair-dependence may have impact on the decreased physical activity and may cause increased dependence in the daily life [43].

The authors’ own study showed that one third of the residents had mild dementia (36.97%) and slightly lower percentage of the participants had cognitive impairment without dementia (27.27%). Prospective studies showed that cognitive impairment is a strong and consistent risk factor for physical disability [44]. Furthermore, the study of Park et al. demonstrated that lower MMSE scores and the intake of neuroleptics make and independent predictors of previous institutionalization [45]. In the systemic review Snowden et al. analyzed the occurrence of disability and chronic diseases in elderly people having various chronic diseases with and without cognitive impairment [46]. The authors demonstrated that decreased cognitive functions have negative impact on functional fitness, increases the scope of institutionalization and mortality rate [46].

The results of the study show that 55% of elderly people covered by the study had experienced moderate depression. Clinical trials evidence that elderly people who have experienced various chronic diseases and disability suffered from depression more often [47]. Dragest et al. observed that depression is associated with poor functioning in the daily life of the NH residents [17]. The authors recommend close observation of the seniors for the depression symptoms and screening with the use of research instrument designed for elderly people [17]. Because of their chronic diseases and very common social isolation caused by mobile disability, elderly people in NHs are exposed to depressive episodes more often than the seniors living with their families [48].

The results of research showed that the average daily functioning score (BI) was evidenced at the level of 54.36 points, where 68.48% of the study participants had severe dependence and over one third of them had moderate dependence. Our analysis proved significantly lower scores of BI for women (*p* = 0.027). This corresponds with results obtained by Ćwirlej-Sozańska et al. who demonstrated higher level of functional disability among older women [49]. The studies investigating gender differences in 11 European countries, the United States and the United Kingdom, showed that women were more often exposed to disability and depression than men [50]. At the time of admission to the NHs most residents required support in doing everyday activities, and the dependence increased in the next years [51]. During two-year observation of the elderly people under institutional care Jerez-Roig et al. observed that the process of the decreasing daily functioning was much quicker at the beginning of the institutionalized care than it was at the end of the study [52]. The research shows that the level of functional disability and dependence of elderly people in everyday activities is much higher in residents of NHs as compared with the results of people living in the community [53]. Taking into account the above-mentioned results we need to propose a systemic assessment of elderly people which can be modelled on the Comprehensive Geriatric Assessment [54].

Decreased daily functioning, defined with the use of fitness tests assessing manual dexterity, strength, endurance and flexibility of upper limbs, as well as body balance and lung capacity was observed in the test sample. The results of our study deviated from the standards for elderly people as proposed by Rikli and Jones [55]. Additionally, Heiland et al. in their population-based study demonstrated that decreased mobility and imbalance were associated with the increased disability risk, irrespective of chronic diseases and cognitive state [56]. Interventions aimed at improving physical fitness may indirectly facilitate social interactions and improve mental state by increasing the mobility and enhancing functional skills [57].

The results of our study showed significant, positive correlations between cognitive functioning and BBS, BBT, BS, ACT, PEF, the range of the shoulder joint flexion, shoulder extension and abduction. In addition, we proved that cognitive functioning was positively correlated with shoulder abduction for women, with PEF and shoulder flexion and extension for men and with BBS, BBT, HGS and ACT for both women and men. This finding may suggest that maintaining functional ability affects seniors cognitive functioning and enables greater independence in their lifestyle. Moreover, the balance maintaining is controlled by a number of physiological systems. Diminishing of neuro-musculoskeletal system functions may increase the risk of fall [58]. Clouston et al. prepared a systemic review in order to establish the correlation between indicators of the changes observed as the ageing progresses, and the physical and cognitive functioning of elderly people [59]. The researchers claim that physical functioning was significantly correlated with the changes of cognitive functions [59]. They also suggest that the assessment of the level of physical and cognitive functioning may help the clinicians and researchers to better identify the risks among elderly people [59]. It is commonly known fact that in general men manifest greater muscle strenght than women. This was confirmed in our research [60]. However, beside strength, it should be equally important for seniors to work over balance and shoulder mobility. Many residents of NHs require walking assistance devices and aids to perform their daily activities. The necessity to use upper limbs to perform such activities as standing up, dressing oneself, taking care of personal hygiene is associated not only with an appropriate strength and flexibility of the upper body but also with the involvement of cognitive functions [61].

The authors’ own study evidences that the persons who obtained higher GDS-15 scores earned lower BBS, BBT, ACT scores and they were found to demonstrate lower range of flexibility of the shoulder joint. The negative correlation between depression and respectively BI, BBS, BBT and HGS for women and BI, BBS and BP for men was determined. This means that depression has negative impact on the physical fitness in seniors using wheelchairs. Regardless the sex, long-term depression symptoms like tiredness and pain cause physical disability occurrence [62]. The study results of Kaup et al. show correlation between depression and decreased physical fitness as well as difficulties in daily living in the older NH residents [63]. During a yearlong observation Tinetti et al. confirmed that depression and general decrease of daily functioning (decreased physical fitness of upper and lower limbs, decreased strength of upper limbs) was associated with increased risk of falling, urinary incontinence and the dependence on other people in daily living [64]. In spite of the existing evidence confirming positive correlation between physical and mental aspects of health there are few studies which cover elderly wheelchair users who live in NHs.

Furthermore, studies showed that the quality of daily functioning was positively correlated with high scores in BBS, BBT, body flexibility, muscle strength and endurance, shoulder flexion. Konopack et al. proved that sex significantly correlates with certain physical fitness aspects. Men comparing to women manifest greater physical strenght within both upper and lower body while the latter characterize with higher body flexibility [65]. Alexandre et al. found that seniors in early phase of disability process experienced difficulties with dressing which requires upper limb strenght, upper and lower body flexibility [66]. The authors also pay attention to adaptation strategies such as performing activities of daily living in sitting positions and using buttonless clothings or stringless shoes. These strategies help the elderly to increase their independence in everyday living [66]. Chang et al. proved that the wheelchair-dependent seniors with decreased physical fitness are characterized by decreased daily functioning [67]. It needs to be noted, however, that these persons had dementia [67]. Brown et al. claim that general daily functioning is best measured with the use of fitness tests which assess body balance, coordination and muscle strength [68]. The results of their studies showed that daily functioning was significantly associated with the measurement results of strength, body balance and shoulder joint flexibility [68]. On the other hand, Foldvari et al. arrived at contrary results and demonstrated that upper limb muscle strength is not a significant predicator of daily functioning, as opposed to lower limb muscle strength, which may account for the use of a wheelchair [69]. Furthermore, these researchers believe that lower limb function worsens quicker than the upper limb function as the ageing progresses. Despite being encouraged to perform physical activity regularly, many NH residents, especially the wheelchair-dependent seniors, spend most of their time in the sitting or recumbent position, even though they are able to do independent or assisted exercises, which significantly influences the decrease of daily functioning [70].

The results of our study point to a significant correlation between cognitive functioning and BBS, as well as the range of shoulder joint flexion. Wang et al. claim that decreased daily functioning was correlated with the impairment of cognitive functions, while the increased level of physical fitness may defer the risk of senile stupor [71]. According to Lane et al. the connection between imbalance and impairment of cognitive functions is associated with increased disability within two years [51]. Based on our knowledge, it is the first study assessing the correlation between physical and mental health in elderly, wheelchair-dependent people without dementia who live in NHs. Previous studies focused on people with high functional fitness or seniors with senile stupor [43,72]. Additionally, the introduction of physical exercises leads to the improvement of daily functioning and consequently it may have positive impact on the seniors by decreasing the level of fear, stress, social isolation, reducing the symptoms of depression and thus contributing to maintaining mental health [73,74].

The analysis of stepwise regression showed significant relationship between the symptoms of depression and daily functioning. Higher BI and ACT scores were associated with mild symptoms of depression. Drageset et al. showed that there is a relationship between the symptoms of depression and daily functioning, and that this relation decreases as the ageing progresses [48]. Although high rates of depression in elderly people who live in NHs have been confirmed, the results show insufficient diagnosis of symptoms of depression, which leads to significant intensification of disability [75,76]. Yang and George proved correlation between the symptoms of depression and daily functioning. They also emphasized that medical interventions aimed at improving the mental health of the seniors should be oriented at improving the physical fitness in daily life [77]. Furthermore, an inactive lifestyle of elderly people who live in NHs leads not only to disability, but it also causes fatigue, increases anxiety, irritability, mood disorders and depression. Physical activity constitutes one of the factors determining psychophysical health in human beings [78,79].

The results of the author’s own study point to significant connections between the level of daily functioning and BBS, BS, HGS as well as the shoulder flexion. Heiland et al. showed that limitations identified in balance tests were associated with the increased disability in daily life, especially in seniors from the oldest age groups [56]. This connection was independent of chronic diseases and cognitive state [56]. According to Onder et al. the measurements of the general physical fitness, particularly of lower and upper limbs, correlate significantly with increasing disability [80]. Furthermore, Legrand et al. determined that low parameters in the assessment of strength, body balance and flexibility are strong predictors of disability, dependence on caregivers and—consequently—mortality [81].

For the elderly people decreased mobility further decline of body balance, muscle strength, flexibility and range of flexion may imply additionally increase of the wheelchair dependence, or they may even become bedridden.

### Study Limitations

The study was of cross-sectional type. This means that we were not able to clarify the causality. The results contain only single timepoint scores thus we could only define relations between variables. The study was carried out in one selected region of country of Poland hence the authors suggest interpreting presented results with caution.

## 5. Conclusions

The results of the study may serve to develop suitable exercise programs for seniors with reduced mobility living in NHs to prevent or slow down the age-related diminishing of physical and mental health.

Obtained results show that upper body flexibility and hand grip strenght have significant impact on daily functioning among both women and men. Thus, it is important to include stretching and strenght exercises in everyday physical activity.

Moreover, body balance and shoulder flexion affect the cognitive and daily functioning hence activities improving these elements should also be incorporated to the exercise programs. All these efforts are meant to improve seniors’ independence in their everyday living.

## Figures and Tables

**Figure 1 ijerph-17-01706-f001:**
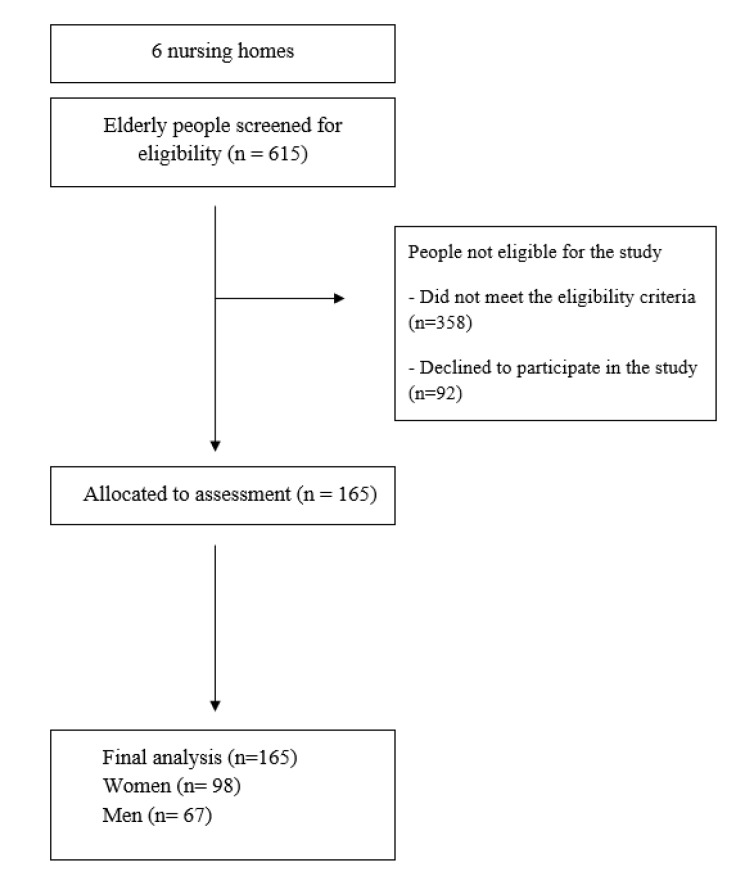
Flow diagram of participants through the study.

**Table 1 ijerph-17-01706-t001:** Sociodemographic and clinical characteristics of the participants.

Variable		Total (*n* = 165)	Women (*n* = 98)	Men (*n* = 67)	*p*-Value
		Number (%)			
Sociodemographic and clinical			Mean (SD)		
Age		74.35 (7.33)	75.87 (7.03)	72.12 (7.25)	0.001 *
BMI (kg/m^2^)		27.02 (4.44)	26.21 (4.40)	28.21 (4.26)	0.004 *
Education					
	Basic	50 (30.3)	29 (29.59)	21 (31.34)	0.171
	Secondary	99 (60.0)	56 (57.14)	43 (64.17)	
	Higher	16 (9.7)	13 (13.26)	3 (4.48)	
Marital status					
	Married	17 (10.3)	9 (9.18)	8 (11.94)	0.007 *
	Widow/Widower	73 (44.2)	54 (55.10)	19 (28.36)	
	Divorced	28 (17.0)	14 (14.29)	14 (20.90)	
	Single	47 (28.5)	21 (21.43)	26 (38.81)	
Length of NH residency					
	>12 moths	14 (8.5)	7 (10.45)	7 (7.14)	0.639
	1–5 years	5 (27.3)	19 (28.36)	26 (26.53)	
	6–10 years	77 (46.7)	32 (47.76)	45 (45.92)	
	over 10 years	29 (17.6)	9 (13.43)	20 (20.41)	
Length of wheelchair dependence					
	>12 months	3 (1.8)	2 (2.04)	1 (1.49)	0.891
	1–5 years	76 (46.1)	46 (46.94)	30 (44.78)	
	6–10 years	72 (43.6)	29 (43.28)	43 (43.87)	
	over 10 years	14 (8.5)	7 (10.45)	7 (7.14)	
Chronic disease					
	Cardiovascular	117 (70.9)	71 (72.45)	46 (68.66)	0.598
	Pulmonary	31 (18.8)	19 (19.39)	12 (17.91)	0.811
	Neurological	113 (68.5)	70 (71.43)	43 (64.18)	0.325
	Urinary system	42 (25.5)	25 (25.51)	17 (25.37)	0.984
	Digestive system	25 (15.2)	18 (18.37)	7 (10.45)	0.164
	Muscoskeletal	83 (50.3)	49 (50.00)	34 (50.75)	0.925
	Ophtamological	58 (35.2)	37 (37.76)	21 (31.34)	0.397
	Otological	52 (31.5)	29 (29.59)	23 (34.33)	0.520
Dominant limb					
	Right	138 (83.6)	81 (82.65)	57 (85.07)	0.680
	Left	27 (16.4)	17 (17.35)	10 (14.93)	0.680
Mental status					
MMSE		24.94 (3.37)	24.70 (3.33)	25.28 (3.42)	0.265
	Normal cognition	59 (35.76)	33 (33.67)	26 (38.81)	0.774
	Cognitive impairment without dementia	45 (27.27)	27 (27.55)	18 (26.87)	
	Moderate dementia	61 (36.97)	38 (38.76)	23 (34.33)	
GDS		5.52 (2.72)	5.67 (2.87)	5.30 (2.49)	0.269
	No depression	73 (44.24)	40 (40.82)	33 (49.25)	0.284
	Moderate depression	92 (55.76)	58 (59.18)	34 (50.75)	
Daily Functioning BI		54.36 (11.78)	52.74(11.11)	56.72 (12.42)	0.027 *
	Complete dependence	0	0	0	
	Severe dependence	113 (68.48)	75 (76.53)	38 (56.72)	0.007 *
	Moderate dependence	52 (31.51)	23 (23.47)	29 (43.28)	
Body balance					
	BBS total	15.52 (5.56)	15.44 (5.53)	15.63 (5.64)	0.895
Body flexibility					
	BS (cm)	−38.85 (22.84)	−37.64 (20.35)	−40.63 (25.43)	0.550
	CSR (cm)	−12.42 (19.54)	−10.86 (15.62)	−14.70 (21.82)	0.459
Muscle strength and endurance					
	ACT (repetitions)	9.94 (5.33)	8.94 (4.81)	11.40 (5.73)	0.005 *
	HGS (kg)	17.18 (15.00)	13.02 (6.35)	23.28 (11.44)	<0.001 *
Joint mobility					
	Shoulder flexion (degree)	131.13 (20.61)	127.66(18.41)	136.19 (22.67)	0.007 *
	Shoulder extension (degree)	43.95 (9.99)	43.91 (10.26)	44.01 (9.67)	0.979
	Shoulder abduction (degree)	128.39 (23.14)	125.32 (24.13)	132.90 (20.98)	0.057
Manual dexterity of the upper limb					
	BBT	26.89(11.15)	24.65 (10.25)	30.16 (11.66)	0.002 *
Lung capacity					
	PEF l/min	169.74 (69.06)	150.22 (49.69)	198.28 (82.64)	<0.001 *
	FEV1 0,01 l	1.26 (0.59)	1.06 (0.42)	1.54 (0.70)	<0.001 *

*- statistical significance, SD, Standard Deviation; BMI, body mass index; GDS, Geriatric Depression Scale; MMSE, Mini-Mental State Examination; BI, Barthel Index; BBS, Berg Balance Scale; BBT, Test Box and Blocks; BS, Back Stretch; CSR, Chair Sit and Reach; HGS, Handgrip Strength; ACT, Arm Curl Test; PEF, Peak expiratory flow, FEV1, forced expiratory volume in 1 s.

**Table 2 ijerph-17-01706-t002:** Correlations between physical and mental health among older adults by sex.

Variable	Cognitive Functioning			Depression			Daily Functioning		
	rho			rho			rho		
	Total (*n* = 165)	Women (*n* = 98)	Men (*n =* 67)	Total (*n* = 165)	Women (*n* = 98)	Men (*n* = 67)	Total (*n* = 165)	Women (*n* = 98)	Men (*n* = 67)
BI	0.263 **	0.136	0.377 **	−0.408 **	−0.443 **	−0.357 **	-	-	-
BBS	0.298 **	0.281 **	0.311 *	−0.261 **	−0.205 *	−0.353 **	0.450 **	0.352 **	0.583 **
BP	0.168 *	0.140	0.231	−0.115	−0.027	−0.269 *	0.431 **	0.331 **	0.571 **
CSR	0.132	0.167	0.087	−0.047	−0.084	0.005	0.314 **	0.207 *	0.477 **
HGS	0.256 **	0.203 *	0.295 *	−0.087	−0.150	−0.031	0.393 **	0.315 **	0.382 **
ACT	0.259 **	0.211 *	0.265 *	−0.318**	−0.353 **	−0.181	0.316 **	0.269 **	0.275 *
Shoulder flexion	0.278 **	0.179	0.384 **	−0.189*	−0.163	−0.132	0.395 **	0.240 *	0.480 **
Shoulder extension	0.12	0.021	0.270 *	−0.10	−0.106	−0.089	0.310 **	0.342 **	0.276 *
Shoulder abduction	0.221 **	0.228 *	0.167	−0.13	−0.146	−0.032	0.203 **	0.071	0.269 *
BBT	0.282 **	0.225 *	0.252 *	−0.237 **	−0.224 *	−0.130	0.380 **	0.227 *	0.435 **
PEF l/min	0.156 *	0.056	0.261 *	−0.15	−0.095	−0.179	0.14	0.113	0.089
FEV1(l)	0.12	0.035	0.126	−0.11	−0.107	−0.038	0.186 *	0.091	0.211

* *p* < 0.05; ** *p* < 0.01; BI, Barthel Index; BBS, Berg Balance Scale; BBT, Test Box and Blocks; BS, Back Stretch; CSR, Chair Sit and Reach; HGS, Handgrip Strength; ACT, Arm Curl Test; PEF, Peak expiratory flow, FEV1, forced expiratory volume in 1 s.

**Table 3 ijerph-17-01706-t003:** Independent variables associated with cognitive functioning, depression and daily functioning of the participants.

Variable		Cognitive Functioning					Depression					Daily Functioning			
		*R*^2^ = 0.114, *R*^2^_adj_ = 0.103, *F* = 10.383					*R*^2^ = 0.191, *R*^2^_adj_ = 0.181, *F* = 21.327					*R*^2^ = 0.348, *R*^2^_adj_ = 0.331, *F* = 19.161			
	β	B	(95 CI)		*p*-Value	β	B	(95 CI)		*p*-Value	β	B	(95 CI)		*p*-Value
Constant	-	18.14	14.84	21.44	<0.001	-	10.82	9.02	12.63	<0.001	-	32.56	20.29	44.84	<0.001
BI	-	-	-	-	-	0.35	0.08	0.16	0.05	<0.001	-	-	-	-	-
BBS	0.22	0.14	0.05	0.23	0.003	-	-	-	-	-	0.29	0.62	0.33	0.91	<0.001
BS	-	-	-	-	-	-	-	-	-	-	0.22	0.11	0.04	0.19	0.003
ACT	-	-	-	-	-	0.18	0.09	0.17	0.02	0.018	-	-	-	-	-
HGS	-	-	-	-	-	-	-	-	-	-	0.17	0.20	0.04	0.37	0.018
Shoulder flexion	0.23	0.04	0.01	0.06	0.004	-	-	-	-	-	0.18	0.10	0.02	0.18	0.015
Women	-	-	-	-	-	-	-	-	-	-	-	-	-	-	-
Men	-	-	-	-	-	-	-	-	-	-	-	-	-	-	-

BI, Barthel Index; BBS, Berg Balance Scale; BS, Back Stretch; CSR, Chair Sit and Reach; HGS, Handgrip Strength; ACT, Arm Curl Test.

## Data Availability

The datasets used and analyzed in the current study are available from the corresponding author on reasonable request.

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
