# Peer review of "Psychophysical Health Factors and Its Correlations in Elderly Wheelchair Users Who Live in Nursing Homes"

_ijerph, 2020, doi:10.3390/ijerph17051706_

Round 1

Reviewer 1 Report

Comments to the Authors

This is an interesting study about psychophysical health condition and its correlations in elderly wheelchair users who live in nursing homes including 165 older wheelchair users.

Materials and Methods:

Page 3, line 97-98: I would include the lack of consent to participate in the study in the exclusion criteria.

Results:

In table 3: I suggest increasing the font, in column 1 the signature of the mentioned value is missing - please complete, no explanation of abbreviations used - please complete, R2 value for models and standardized beta coefficients are not provided. I would complete all the values in the table, even if they were not significant.

Discussion:

Summary of the findings or conclusions and limitations of a study can be added.

Author Response

Dear Reviewer,

We are very thankful for your comments about our article "Psychophysical health condition and its correlations in elderly wheelchair users who live in nursing homes”. We included all your helpful advice and made necessary changes in the article.

The manuscript has been carefully rechecked and appropriate changes have been made in accordance with the reviewers’ comments. Changes to the main text of manuscript according to these comments have been marked in red font. 

Point 1: Materials and Methods:

Page 3, line 97-98: I would include the lack of consent to participate in the study in the exclusion criteria. 

Response 1:

The exclusion criteria include lack of consent of patient or his doctor – point (3) on page 3, line 95. The suitable excerpt from source text is presented below:

The exclusion criteria were as follows: (1) severe and acute cardiovascular, musculoskeletal, or pulmonary illnesses (2) suffering from spinal cord injury with no rehabilitation potential (3) lack of consent of the elderly person or their doctor for participation in the exercises.

Point 2: Results:

In table 3: I suggest increasing the font, in column 1 the signature of the mentioned value is missing - please complete, no explanation of abbreviations used - please complete, R2 value for models and standardized beta coefficients are not provided. I would complete all the values in the table, even if they were not significant. 

Response 2:

Thank you for suggestion. The data was supplemented as suggested by the reviewer.

The Table 3 contains only statistically significant data of each model. Data from three models were combined in one table to maintain clarityof presented informations.

Point 3: Discussion:

Summary of the findings or conclusions and limitations of a study can be added.

Response 3:

According to the reviewer's suggestion, summary of the findings or conclusions and limitations of a study has been redrafted.

We appreciate all your insightful comments and the opportunity to revise our paper. Thank you for taking the time and energy to help us improve the paper. We hope our revision meets your approval.

Yours faithfully,

Authors

Reviewer 2 Report

Thank you for giving me the opportunity to review this interesting manuscript. However, some minor points should be addressed before considering for publication.

Please move the Ethical Approval and Informed Consent into the 2.1 study participants section.

Why authors did not compare women and men values? Maybe this analyses would be interesting. I encourage authors to compere men and women in the most relevant variables. If they did not consider this analysis interesting enough, it should be highlighted in the limitation section.

Review the citation of Nakamura in line 67. It seems that it is not included. Moreover, authors should review all of them since other citation are not properly included.

Author Response

Dear Reviewer,

We are very thankful for your comments about our article “ Psychophysical health condition and its correlations in elderly wheelchair users who live in nursing homes”. We included all your helpful advice and made necessary changes in the article.

The manuscript has been carefully rechecked and appropriate changes have been made in accordance with the reviewers’ comments. Changes to the main text of manuscript according to these comments have been marked in red font.

Point 1:

Please move the Ethical Approval and Informed Consent into the 2.1 study participants section.

Response 1:

According to the reviewer's suggestion, the Ethical Approval and Informed Consent has been moved to the study participants section.

Point 2.

Why authors did not compare women and men values? Maybe this analyses would be interesting. I encourage authors to compere men and women in the most relevant variables. If they did not consider this analysis interesting enough, it should be highlighted in the limitation section.

Response 2:

In response to the reviewer’s comment, we have added analysis including comparison between women and men. Suitable data were included in table 1 and 2 and described in results and discussion sections.

Point 3:

Review the citation of Nakamura in line 67. It seems that it is not included. Moreover, authors should review all of them since other citation are not properly included.

Response 3:

Thank you for noticing the error. The citations were reviewed and corrected by authors.

We appreciate all your insightful comments and the opportunity to revise our paper. Thank you for taking the time and energy to help us improve the paper. We hope our revision meets your approval.

Yours faithfully,

Authors

Reviewer 3 Report

This article presents a cross-sectional descriptive study with the aim of evaluate the variables affecting the psychophysical state of the elderly people using wheelchairs and living in nursing homes.  

The contribution made to know the correlations between the different psychophysical variables can help design and implement future interventions to prevent physical and mental deterioration in people using wheelchairs living in nursing homes.

Therefore, it is valued as suitable for publication, after rectification of the following aspects:

- According to the recommendations of the journal, the abstract should be a total of about 200 words maximum and yet contains 246 words.

- In Figure 1 (Flow diagram of participants through the study), it is based on an eligible sample of 615 subjects, of which 373 did not meet inclusion criteria and 92 declined participation in the study. This would give a result of 150 subjects as a sample of the present study, but nevertheless, the sample is 165 subjects. It is recommended to review this data or, if it is correct, set out more clearly in the flow diagram.

- It is recommended to review reference number 19 (Nakamura et al.). Although it does appear in the bibliography list, this reference does not appear in the text.

- It is also recommended to review the list of references, as there are aspects to improve according to the guidelines of the journal. In particular, in references 2 and 3 for example should not appear “:” after “accessed on”; some journals do not appear in their abbreviated version (example reference number 31 Maryland State Medical Journal), etc.

- The conclusions section can be improved, in greater coherence with the results previously indicated.

I hope that the suggested changes help to improve the quality of the article and that they are well received.

Kind regards,

Author Response

Dear Reviewer,

We are very thankful for your comments about our article “ Psychophysical health condition and its correlations in elderly wheelchair users who live in nursing homes”. We included all your helpful advice and made necessary changes in the article.

The manuscript has been carefully rechecked and appropriate changes have been made in accordance with the reviewers’ comments. Changes to the main text of manuscript according to these comments have been marked in red font. 

Point 1:

- According to the recommendations of the journal, the abstract should be a total of about 200 words maximum and yet contains 246 words.

Response 1:

We have made proper editing and have shortened the abstract section to match the required lenght.

Point 2.

- In Figure 1 (Flow diagram of participants through the study), it is based on an eligible sample of 615 subjects, of which 373 did not meet inclusion criteria and 92 declined participation in the study. This would give a result of 150 subjects as a sample of the present study, but nevertheless, the sample is 165 subjects. It is recommended to review this data or, if it is correct, set out more clearly in the flow diagram.

Response 2:

We are thankful for noticing such an important mistake. The total number of people screened for eligibility was 615. The number of people who did not meet the eligibility criteria was miswritten as 373 where it should be 358. The number of people who declined participation in the study was 92. 165 persons were included in the study.

Point 3.

- It is recommended to review reference number 19 (Nakamura et al.). Although it does appear in the bibliography list, this reference does not appear in the text.

Response 3:

Thank you for noticing the error. The citations were reviewed and corrected by authors

Point 4:

- It is also recommended to review the list of references, as there are aspects to improve according to the guidelines of the journal. In particular, in references 2 and 3 for example should not appear “:” after “accessed on”; some journals do not appear in their abbreviated version (example reference number 31 Maryland State Medical Journal), etc.

Response 4:

Thank you for noticing the error. The references were reviewed and corrected by authors.

Point 5:

- The conclusions section can be improved, in greater coherence with the results previously indicated.

Response 5:

We agree with the reviewer's opinion. The conclusions section was reedited and redrafted.

We appreciate all your insightful comments and the opportunity to revise our paper. Thank you for taking the time and energy to help us improve the paper. We hope our revision meets your approval.

Yours faithfully,

Authors